# Ancient origin of fucosylated xyloglucan in charophycean green algae

Maria Dalgaard Mikkelsen[1,2], Jesper Harholt [3], Bjørge Westereng [4], David Domozych[5], Stephen C. Fry[6], Ida Elisabeth Johansen[1], Jonatan U. Fangel[1], Mateusz Łężyk[7], Tao Feng[7], Louise Nancke[1], Jørn D. Mikkelsen[7], William G. T. Willats[1,8] & Peter Ulvskov [1✉]

The charophycean green algae (CGA or basal streptophytes) are of particular evolutionary significance because their ancestors gave rise to land plants. One outstanding feature of these algae is that their cell walls exhibit remarkable similarities to those of land plants. Xyloglucan (XyG) is a major structural component of the cell walls of most land plants and was originally thought to be absent in CGA. This study presents evidence that XyG evolved in the CGA. This is based on a) the identification of orthologs of the genetic machinery to produce XyG, b) the identification of XyG in a range of CGA and, c) the structural elucidation of XyG, including uronic acid-containing XyG, in selected CGA. Most notably, XyG fucosylation, a feature considered as a late evolutionary elaboration of the basic XyG structure and orthologs to the corresponding biosynthetic enzymes are shown to be present in *Mesotaenium caldariorum*.

[1] Department of Plant and Environmental Sciences, University of Copenhagen, Copenhagen, Denmark. [2] Department of Biotechnology and Biomedicine, DTU Bioengineering, Technical University of Denmark, Copenhagen, Denmark. [3] Carlsberg Research Laboratory, Copenhagen, Denmark. [4] Department of Chemistry, Biotechnology and Food Science, Norwegian University of Life Sciences, Oslo, Norway. [5] Department of Biology and Skidmore Microscopy Imaging Center, Skidmore College, New York, USA. [6] Edinburgh Cell Wall Group, Institute of Molecular Plant Sciences, The University of Edinburgh, Edinburgh, UK. [7] Department of Chemical and Biochemical Engineering, Technical University of Denmark, Copenhagen, Denmark. [8] School of Natural and Environmental Sciences, Newcastle University, Newcastle upon Tyne, UK. ✉email: Ulvskov@plen.ku.dk

The group of green algae that is most closely related to land plants is the charophycean green algae (CGA or basal streptophytes). Phylogenetic analyses also indicate that land plants (or embryophytes) evolved from an ancestral taxon of the CGA class, Zygnematophyceae[1–6] approximately 500+ million years ago[7]. The subsequent global radiation of land plants profoundly changed the biogeochemistry and natural history of the planet, and has consequently resulted in Earth's highly diverse terrestrial ecosystems[8].

The ancestral CGA that colonized land evolved traits that enabled them to survive in a terrestrial habitat, including specialized cell wall features[9]. This supposition has been supported by recent genomic sequencing of CGA leading to significant insights into the evolutionary origin of many land plant characteristics[5,10].

Extant CGA produce many cell wall polymers that are also found in land plants[11–13]. This has been recently corroborated by establishing orthology between genes involved in the biosynthesis of cell wall polymers in flowering plants and various CGA taxa[5,14,15]. One such polymer type is xyloglucan (XyG)[16]. XyG, a major wall component, has been well-characterized in non-commelinid angiosperm primary cell walls, also referred to as Type-I cell walls[17]. It has been hypothesized that XyG exhibits a close physical affinity for cellulose, forming tethers between microfibrils and, in turn, contributing to the load-bearing capacity of the wall[18]. While this model has been challenged[19–21], the significance of XyG in cell expansion and cell wall architecture is generally accepted. XyG consists of a β-(1,4)-glucan (designated with the one-letter code G) backbone, with sidechains starting with α-(1,6) linked xylose (X). The xylose can be further decorated with different glycosyl groups, such as β-(1,2) linked galactose (L) followed by α-(1,2) linked fucose (F) or β-(1,2) glucuronic acid (Y) followed by β-(1,4) galactose (P)[22–24]. XyG is synthesized in the Golgi apparatus via the action of glycosyl transferases derived from different gene families, several of which have been verified through functional characterization[16,25,26].

Due to the absence of biochemical evidence for XyG being present in CGA, a prominent hypothesis is still that XyG originated in land plants[12]. Furthermore, fucosylation of XyG was thought to have arisen in the common ancestor of flowering plants based on its absence in the cell walls of early divergent land plants[27]. Fucosylated XyG has previously been considered a hallmark feature of seed plants (spermatophytes). However, fucosylated XyG has been found in hornworts[28], while XyG-like glucan and xylosyl linkages have been detected in *Spirogyra* sp., implying that XyG may be present in the CGA[13]. This was supported by epitope-binding studies using XyG-specific monoclonal antibodies[13,29,30], and studies of the biochemical activity of endotransglucosylases and endotransglucosylases/ hydrolases[27,31,32] in various CGA. In this study, we present unambiguous identification of XyG, including fucosylated sidechains, in CGA and infer a path through evolution of XyG in which reduction of structural complexity is more prevalent than further elaboration of the XyG sidechain repertoire.

## Results

**Phylogeny of XyG biosynthesis.** In the green plant lineage (Fig. 1) cellulose is nearly ubiquitous. Recent gene sequencing of several CGA taxa[5,10,33–36] has allowed the identification of CGA orthologs to the cellulose synthase-like C (CslC) clade of CAZy glycosyl transferase family 2 (GT2), containing the Glc β-1-4 XyG-backbone synthase. On the basis of these studies, we have identified sequences from *Mesotaenium, Spirogloea*, and *Spirogyra* that fall squarely in the CslC-clade, see Fig. 2 where sequences 62–63 are the *Spirogloea* genes. This suggests that CGA synthesize

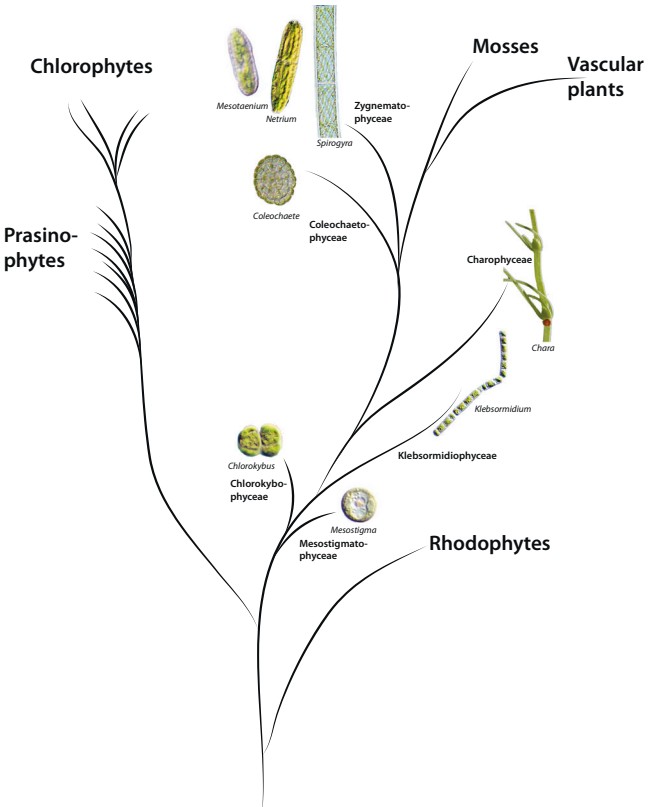

**Fig. 1 Phylogeny of the green plant lineage.** Three later diverged CGA classes are situated as the closest living relatives to land plants; the Charophyceae, Coleochaetophyceae and the Zygnematophyceae, while the Klebsomidophyceae, Chlorokybophyceae and Mesostigmatophyceae are classes of earlier evolved CGA. The Chlorophytes and Prasinophytes are ancestral to CGA. Green algal species of interest here are illustrated by pictures. The Rhodophytes are used as ancestral group for the green plants.

XyG backbone using homologous enzymes to those of land plants. Orthologs required for sidechain biosynthesis (GT34, 37, and 47), were also identified, suggesting that late divergent CGA have a full biosynthetic inventory for production of complex XyGs. XyG sidechain synthesis is initiated by xylosylation by xyloglucan xylosyltransferases (XXTs) belonging to the GT34 family (Fig. S1)[37–39]. The GT34 B-clade comprising the galacto-mannan galactosyltransferases is very poorly resolved, and it will require analyses with more species included to ascertain whether the A and C-clades[37,40] are stable. *Physcomitrella* does not have sequences in either, yet is able to make the L-chain and more elaborate XyG sidechains (but not the F-chain)[28]. *Selaginella* is not represented in the C-clade but features two sequences in the A-clade and is able to make the F-chain[40]. Similar sequences are identified in *Mesotaenium, Spirogyra*, and *Cylindrocystis* whereas *Spirogloea* sequence 80 is on the unresolved borderline between the clades and 78–79 squarely in clade C (Fig. S1).

GTs of GT47 clade A adds the next sugar moiety, either galactose or arabinose (D-chain)[24,41,42]. To investigate the genetic potential of CGA galactosyl transferase capability we cloned two full-length *Coleochaete orbicularis* sequences, CoGT47A1 and 2 (Genbank accession numbers: MW149251 and MW149252, respectively), which are positioned in a subclade subtending the clade of *Arabidopsis thaliana* XLT2 (a galactosyltransferase) and tomato XST1&2 (arabinosyltransferases) (Fig. S2). The *Penium margaritaceum* genome was published[5] after the work for this paper was completed, and is not included in our analysis except

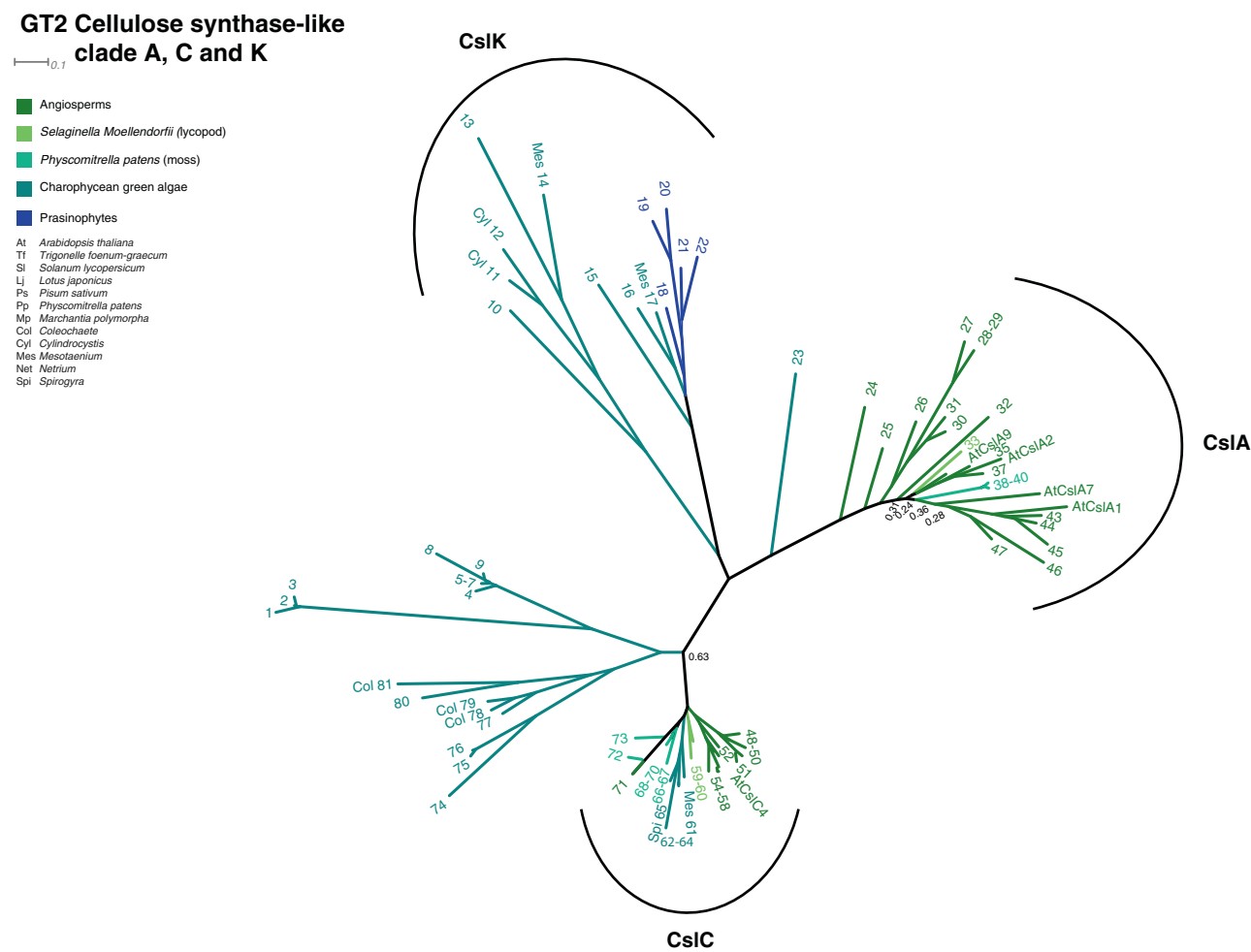

**Fig. 2 Phylogenetic tree of GT2 cellulose synthase-like clade A, C and K.** Branches are color-coded according to taxonomic group. Bootstrap values below 0.7 are indicated along the backbone of the tree. Keys to the numbered sequences are found in supplementary data 1. The numbers are preceded by a three-letter genus prefix whenever a species belonging to the genus has been selected for MALDI-ToF analysis or have been mentioned in the text. Key to the genera is shown to the left in the Figure. Sequences for which experimental evidence of function exist are indicated with their gene names: *Arabidopsis thaliana* AtCslC4 encodes a XyG backbone synthase[77] and AtCslA1, 2, 7, and 9 are glucomannan backbone synthases[78].

that Supplementary data 1 provides a list of the genomic sequences of the GT47 A-clade. We note that no *P. margaritaceum* sequence is closely related to MUR3/Katamari and that *P. margaritaceum* sequences with the highest identity to XLT2 are positioned in the same subtending branch as the *Mesotaenium* Mes98-99 (Supplementary data 1 and Fig. S2).

GT37, the family comprising the FUT fucosylating the L-chain in *A. thaliana*[43–45], warrants a more detailed discussion. Fucosylation of an Ara*p* in a D-chain (turning it into an E-chain)[24] has been observed in early diverging vascular plants, *Microsorum, Equisetum, Selaginella*[40,46]. This observation, as well as the notion that fucosylation by GT37 members is regio-specific lends support to the hypothesis that a clade in GT37 containing *Physcomitrella patens* and *Selaginella moellendorffii* sequences, that cluster separately to later diverging land plants, comprises the activity required for D-chain fucosylation. Several GT37-type encoding sequences have been found amongst CGA and these cluster in the putative D-chain fucosyltransferase clade (Fig. S3). Supplemental data 1 provides keys to all trees.

**Detection in extracts using XyG-specific monoclonal antibodies.** Over 40 different CGA species from all classes of CGA were probed for XyG occurrence by comprehensive microarray polymer profiling (CoMPP)[47], using two different XyG specific

antibodies LM15 and LM25, see heatmaps in Fig. 3 and S4. The early divergent green algae *Chlorokybus* and *Klebsormidium* did not show any labeling, whereas a low xyloglucanase (E-XEGP) digestible signal was observed in *Mesostigma viride* (Fig. 3 and S4). Positive labeling was observed in many species of the late divergent CGA. The XyG recognized by LM25 was susceptible to E-XEGP treatment, whereas LM15 specific XyG showed an intriguing recalcitrance to E-XEGP treatment, normally not observed in higher plants[48]. Interestingly, XyG labeling and E-XEGP susceptibility varied between different species and isolates of the same genus (Fig. 3 and S4). The recalcitrance towards xyloglucanase was further investigated using a xyloglucanase (XcXGHA)[48] with a unique specificity towards substituted XyG, cleaving between two X-chains, which is in contrast to the commonly used xyloglucanase, which cleaves between a substituted and a non-substituted glucosyl residue in the xyloglucan backbone. The results, however, showed a similar recalcitrance, suggesting that is likely not caused by a lack of un-substituted residues (Fig. S5). The specificities of LM15 and LM25 are not identical and neither are their sensitivities, with LM25 being substantially more sensitive. LM15 was raised against an XXXG conjugate and cross-reacts to some extent with XLXG, while LM25 was raised against an XLXG/XLLG conjugate[49] and cross-reacts to some extent with XXXG. We suggest that LM25 detects

| Antibody | LM25 | | | | LM15 | | | |
|---|---|---|---|---|---|---|---|---|
| Extraction | CDTA | | NaOH | | CDTA | | NaOH | |
| Enzyme treatment | - | E-XEGP | - | E-XEGP | - | E-XEGP | - | E-XEGP |
| *Mesostigma viride* | 0 | 0 | 15 | 0 | 0 | 0 | 0 | 0 |
| *Chlorokybus atmophyticus* | 0 | 0 | 0 | 0 | 0 | 0 | 0 | 0 |
| *Entransia sp.* | 12 | 0 | 18 | 5 | 0 | 0 | 0 | 0 |
| *Klebsormidium sp.* | 0 | 0 | 7 | 0 | 0 | 0 | 0 | 0 |
| *Klebsormidium flaccidum* | 0 | 0 | 0 | 0 | 0 | 0 | 0 | 0 |
| *Klebsormidium dissectum* | 0 | 0 | 0 | 0 | 0 | 0 | 0 | 0 |
| *Chara corallina* | 0 | 0 | 11 | 0 | 26 | 22 | 19 | 20 |
| *Coleochaete nitellarum* | 20 | 12 | 100 | 74 | 25 | 24 | 26 | 22 |
| *Coleochaete orbicularis* | 0 | 0 | 33 | 5 | 0 | 0 | 13 | 0 |
| *Closterium acerosum* | 0 | 0 | 10 | 0 | 0 | 0 | 0 | 0 |
| *Cosmarium turpini* | 11 | 0 | 37 | 19 | 30 | 30 | 38 | 39 |
| *Micrasterias furcata* | 0 | 0 | 13 | 0 | 0 | 0 | 0 | 0 |
| *Pleurotaenium trabecula* | 0 | 0 | 0 | 0 | 0 | 0 | 0 | 0 |
| *Teilingia granulata* | 0 | 0 | 0 | 0 | 0 | 0 | 0 | 0 |
| *Tetmemorus brebissonii* | 10 | 0 | 15 | 6 | 0 | 0 | 0 | 0 |
| *Cylindrocystis brebissionii* | 11 | 0 | 57 | 0 | 0 | 0 | 8 | 0 |
| *Mesotaenium caldariorum* | 0 | 0 | 67 | 0 | 31 | 29 | 62 | 52 |
| *Netrium interruptum* | 6 | 0 | 37 | 0 | 25 | 24 | 46 | 7 |
| *Penium margaritaceum* | 6 | 0 | 19 | 6 | 0 | 0 | 0 | 0 |
| *Mougeotia transeau* | 0 | 0 | 0 | 0 | 51 | 54 | 56 | 57 |
| *Spirogyra communis* | 7 | 0 | 42 | 23 | 50 | 52 | 48 | 53 |

**Fig. 3 Heatmap of Comprehensive Microarray Polymer Profiling (CoMPP) of a range of CGA species.** Samples were sequentially extracted with CDTA and NaOH. Spotted arrays were probed with XyG-specific antibodies LM15 and LM25 with or without pre-digestion with-XEGP xyloglucanase.

a separate and smaller XyG population and there are a number of reasons why catalytically active xyloglucanases may cleave XyG yet only reduce LM15 binding to a minor extent. If the product oligosaccharides are sufficiently large, they will likely remain immobilized to the nitrocellulose. Similarly, if they are covalently attached to other macromolecules they may remain immobilized. In this regard, it is worth-noting that endo-transglycosidases in CGAs are rather donor substrate tolerant and can graft mannan, for example, onto XyG[50]. It is also possible that certain levels and/or patterns of substitution may reduce the accessibility of a xyloglucanase to, or its activity against, a portion of XyG that nonetheless remains detectible by LM15. This effect has been noted previously for other substrate/enzyme/mAb systems. For example, Willats et al.[51] reported that despite extensive degradation of pectin by a highly active endo-polygalacturonase, there was a residue of recalcitrant homogalacturonan (HG) that was still detectable by anti-HG mAbs and this was most likely due to differential branching patterns[51].

The defining substructure of XyG is X, i.e., the disaccharide isoprimeverose and the release of isoprimeverose by Driselase™ digestion is thus diagnostic of XyG[52]. *Mesotaenium caldariorum* and *Netrium digitus* were subjected to Driselase digestion and isoprimeverose was detected, although it was not detectable in *Coleochaete orbicularis* (Fig. S6a), nor had it been detected earlier by identical methods in *Coleochaete scutata*, *Chara* and *Klebsormidium*[12], indicating that it is not a quantitatively major component of CGA polysaccharides[12]. Based on these results and our labeling studies (see below) we propose that XyG is developmentally dynamic in the cell walls of the CGA studied,

possibly, due to involvement in cell division and wall expansion, and that this may be the reason for the variable results.

**Immunohistochemistry**. In order to identify the cellular localization of XyG in CGA, immunolabeling with the XyG-binding LM15 monoclonal antibody, was performed on live and fixed/resin-embedded algae (Fig. 4 and Figs. S7–S9). XyG epitopes were detected in all species investigated. Labeling was found within specific cell domains, especially in expanding or dividing cells (Fig. 4 and Figs. S7–S9). To further explore the possible role of XyG in cell division or cell expansion, synchronized *Mesotaenium caldariorum* cells were labeled with LM15 (Fig. 5). Labeling was noted from the beginning of the production of the new "primary cell wall" at the isthmus, i.e., the site of cell expansion and division (Fig. 5a, b). It should be noted that this is not analogous to "primary cell walls" of land plants. Daughter cells retained labeling in the zones near the division zone (Fig. 5c–f). After the daughter cells stopped expanding, the primary wall was shed, which is a typical wall developmental event in *Mesotaenium* species[53] (Fig. 5g–h). These results indicate that XyG, as highlighted by LM15, is produced in expanding walls during semi-cell morphogenesis but not in non-expanding mature cell walls.

The physical masking of a cell wall polymers including XyG by other polymers may prevent antibody labeling which has been observed in several immunocytochemical studies. Previous investigations that employed enzyme-based "unmasking" of wall polymers followed by immunolabeling in higher plants[54] have provided insight into this phenomenon and yielded practical protocols to label wall polymers. Our study focused on chemical

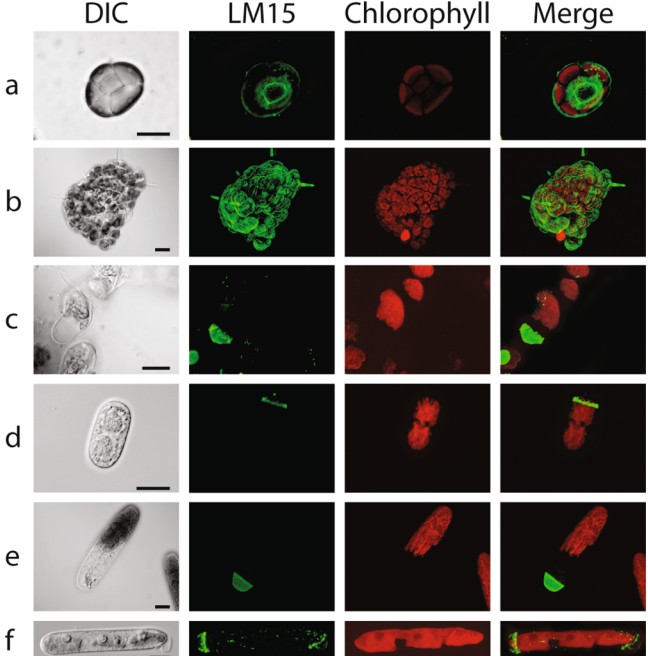

|  | DIC | LM15 | Chlorophyll | Merge |
|---|---|---|---|---|

a
b
c
d
e
f

**Fig. 4 XyG Immunolabeling localizes to the cell wall of several CGA species from the three later evolved classes of CGA.** **a** *Coleochaete orbicularis* XyG LM15 labeling is strong in the remnants of the zoospore cell wall and in the periphery of the newly formed thallus. **b** The peripheral cells of thalli as well as elongating hairs also show distinct labeling. **c** *Coleochaete nitellarum* XyG LM15 labeling is strong at the tips of growing filament structures. In elongating **d** *Cylindrocystis brebissonii*, **e** *Netrium digitus* and **f** *Mesotaenium caldariorum* cells, XyG specific labeling with LM15 is restricted to zones in the cell wall near one end or both ends of the cell, likely involved in cell expansion. DIC: light image, LM15: TRITC (green), chlorophyll autofluorescence (red). Merge: LM15 and chlorophyll autofluorescence. Scalebars as indicated in the DIC images: **a** 10 μm and **b**–**f** 20 μm.

composition not architecture, and we did not pursue investigations of possible polymer masking. It is important to note that our in vivo labeling depicts epitopes that are accessible to the antibody. However, uniform labeling of the walls was never observed, and this lack of labeling in mature walls may be the result of incorporation of XyG into wall constructs that render it inaccessible to the antibody. Likewise, it may be the result of XyG metabolism in different developmental stages. While these two scenarios are not mutually exclusive and labeling appears to correlate with division and/or expansion, future work using systematic unmasking protocols is required to elucidate the presence of XyG at different developmental stages.

**XyG fine structure mapping using MALDI-ToF.** We selected CGA taxa representing the late divergent CGA classes, Coleochaetophyceae and Zygnematophyceae, which showed high labeling and substantial E-XEGP susceptibility in COMPP and analyzed XyG composition in their cell walls using MALDI-ToF (OLIMP)[55] (Fig. 6). Peaks matching known XyG oligomers (XGO) and with mass differences in series of related XGOs, hexoses (*m/z* of 162), pentoses (*m/z* 132), uronic acids (*m/z* 176), and deoxy-hexoses (*m/z* 146) could be observed in all species analyzed. In *Coleochaete orbicularis* XGOs with mass differences corresponding to hexoses and xyloses were found, strongly suggesting the presence of non-fucosylated XyG (Fig. 6). In *Cylindrocystis brebissonii m/z* peaks corresponding to XGOs could be identified corresponding to differences of uronic acids (Fig. 6). XyG with galacturonic acid (GalA) has previously been found in

root hairs of *A. thaliana*[56] and in *P. patens* and *Marchantia polymorpha*[28], while glucuronic acid has never been found in XyG to date. *Coleochaete orbicularis* and *Cylindrocystis brebissonii* feature sparsely branched XyG, with the suggested structures of LGGGG (1146), PGGGG (1322), LLGGG (1440), and LPGGG (1616), forming a new series of XGOs (Figs. 6 and 7). Furthermore, there were indications of a XyG specific acetylation (1616->1658 a mass difference of 42), although this needs further investigation. In *Netrium digitus* the *m/z* differences found correspond to hexose, pentose and uronic acid (Fig. 6), suggesting XyG with a complex decoration with both hexoses, pentoses and uronic acids, but generally with smaller XGO masses.

The yield of enzymatically extractable XGOs in *Mesotaenium caldariorum* was high compared to the yield from the other species. The *m/z* differences implied the presence of hexoses, pentoses, uronic acids, and very interestingly also deoxy-hexoses, (Fig. 6) identified as fucose, see below. The latter have, to date, only been identified in land plants. XyG structure can vary substantially between tissue and cell types as observed in a study of monocots[57], of fucosylated XyG in tobacco[58] and of XyG featuring uronic acid XyG[28,54,56–58]. Fucosylated XyG, featuring the F-chain, is the dominating type in *A. thaliana* while the galacturonic acid (GalA) containing the Y sidechain (Fig. 7a) has been found only in root hairs[56].

To verify the findings of fucosylated and galactosylated XyG in *Mesotaenium caldariorum*, xyloglucanase digestion was supplemented with either α-1-6-galactosidase (E-BGLAN, Megazyme) or α-L-fucosidase (MFuc5)[59]. After fucosidase treatment, *m/z* 1099 and 1393 disappeared, while *m/z* 1555 was partially degraded (Figs. 7c and 8c). The predicted *m/z* after degradation of these peaks was 953, 1247, and 1409, respectively. These *m/z*'s where already present before fucosidase treatment, e.g., no new masses were produced, although the peaks all appeared to increase in relative abundance. The *m/z* of these three fucosidase degradable XGOs corresponds to the masses of XFG, XXFG, and XLFG, respectively. The presence of XXFG was further supported by TLC analysis where a distinct band co-migrated with the standard XXFG (Fig. S6b), though the corresponding band was very faint in *Netrium digitus*, undetectable in *Coleochaete orbicularis* (Fig. S6b) and had not been detected by identical methods in *Chara*[12]. In addition, CoMPP analysis using antibodies specific for fucosylated XyG, CCRC-M1, and CCRC-M39, revealed antibody labeling in *Mesotaenium caldariorum* which was diminished after treatment with fucosidase. Labeling of fucosylated XyG was undetected in *Netrium digitus*, *Coleochaete orbicularis* and *Cylindrocystis brebissonii* supporting the MALDI-ToF analysis (Fig. S10).

Based on the MALDI-ToF analysis, galactosylated XGOs are abundant in *Mesotaenium caldariorum* and galactosidase digestion shows that *m/z* 953, 1115, 1247, 1409, and 1555 were susceptible to galactosidase digestion, suggesting that the peaks are XLG, GXLG/GLXG/LLG, XXLG/XLXG, XLLG, and XLFG, respectively (Fig. 8b). Some of these findings were also supported by TLC analysis, where XGO co-migrated with XXLG/XLXG and another XGO possibly co-migrating with XLLG (Fig. S6b). Additionally, it was observed that the series of related XGOs with *m/z* of 1129, 1423, 1445, and 1585 were susceptible to galactosidase degradation, resulting in the appearance of *m/z* of 967, 1261, and a less abundant *m/z* of 1283 (Fig. 8b). Tandem MS, performed on the XGOs *m/z* 1129 and 1423 (Fig. 8d, e), both of which are galactosidase sensitive, revealed that they differ by an extra X in 1423, 1129 comprising one terminal pentose, a terminal hexose (likely the terminal galactose, determined by galactosidase treatment), and a terminal uronic acid. These findings suggest the presence of sidechain P. The branching order in these XGOs is generally unknown so possible structures

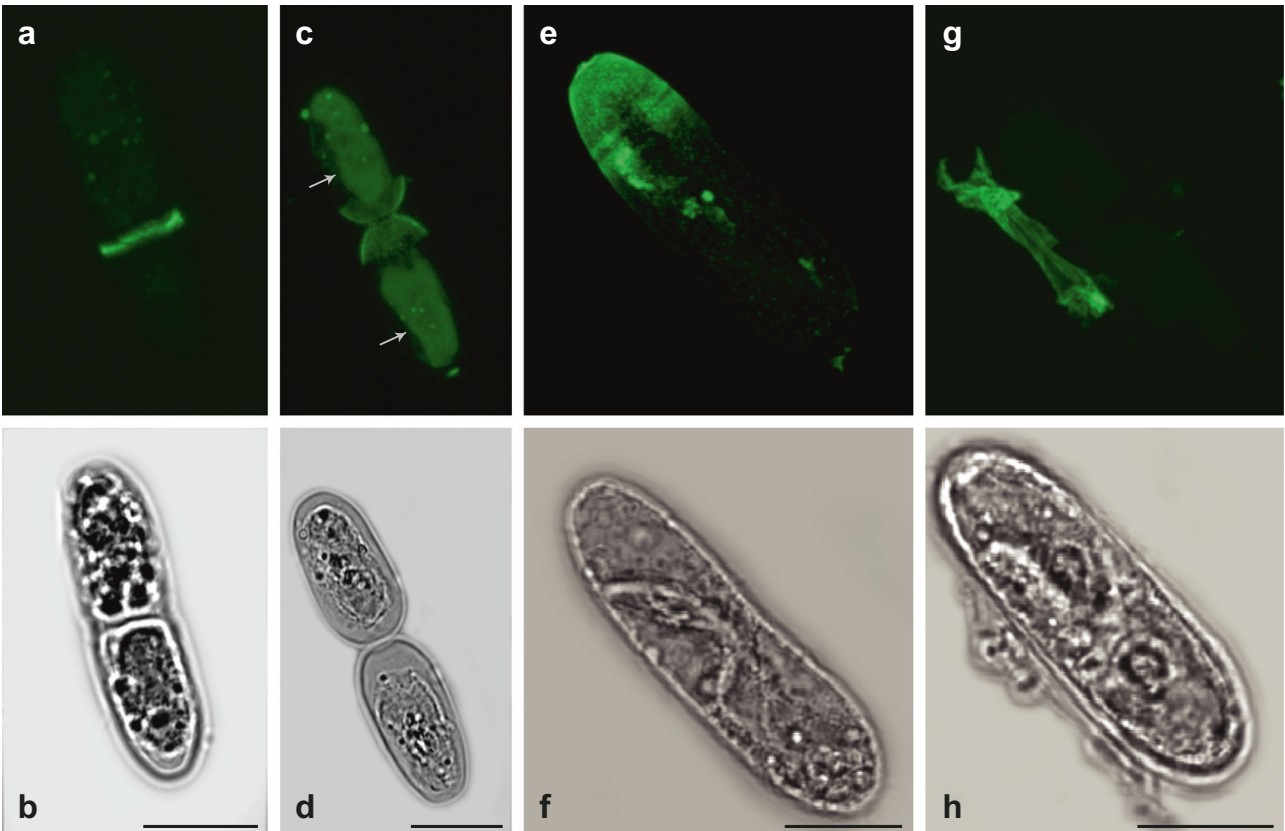

**Fig. 5 XyG production in *Mesotaenium caldariorum* during semicell morphogenesis.** Immunolabelling of cell cycle synchronized *Mesotaenium caldariorum* cells during semicell morphogenesis. **a, b** LM15 binding was found in the isthmus when cells were starting to divide. **c, d** LM15 is evident in the ends of daughter cells near the division zone (while chlorophyll is also evident and colored green in the middle of the two daughter cells as indicated by arrows) **e, f** cell expansion is almost completed of the new *Mesotaenium caldariorum* cells and LM15 labeling is very clear in the newly formed "primary cell wall". **g, h** Cell division and expansion is completed and the "primary cell wall" is cast off as a single piece depicted here, including the LM15 labeling. Top panel: LM15 (TRITC) (green), lower panel: DIC light image. **c** Chlorophyll autofluorescence (green), in the middle of the cells. Scalebars as indicated in the DIC images: 20 μm.

comprise XPG/PXG and XXPG/XPXG/PXXG, respectively. BPGG is an alternative structure of XXPG shown in Fig. 8b. We consider this unlikely as the XGOs of Fig. 8 would then no longer belong to the same series.

Taken together, these data show that *Mesotaenium caldariorum* produces XyG featuring sidechains previously thought to have evolved in land plants. The emergence of XyG fucosylation among later-evolving CGA raises the question of how early the origin of XyG itself should be sought. We searched the *Chlorokybus* genome for sequences related to ClsC or CslK that putatively could encode XyG backbone synthases and found none. However, we found a CESA gene that appears to fulfill the criteria for being rosette forming[60] (Fig. S11), suggesting that plant cell wall biosynthesis, as we know it from land plant studies, has its origin in a common ancestor of *Chlorokybus* and land plants[10].

## Discussion
The discussion of XyG sidechains presented by Schultink et al.[24], suggests that the evolution of XyG structure is one of gradual simplification, such that X-, L-, and F-chains eventually become the predominant sidechains. It is thus not surprising that the repertoires of XyG sidechains vary within the CGA, with the observation of a new series of negatively charged and sparsely branched XGOs in *Coleochaete* and *Cylindrocystis* as striking examples. Here, we extend our understanding of the evolutionary history of XyG biosynthetic machinery[15,61], and propose that the

origin of XyG roughly coincided with the emergence of the phragmoplast. Furthermore, we contribute the identification of CGA Csl sequences that are unambiguously assigned to the CslC clade and provide *Coleochaete* full-length sequences orthologous to XLT2 and XST1&2 (GT47, Fig. S2).

The P and Q sidechains, which are unique in containing galactosyl bound to the *O*-4 position together with a galacturonic acid linked to *O*-2 on the same xylose, have been identified in liverworts and mosses[28] and not in hornworts, whose cell walls feature GalA-containing XyG motifs that are indistinguishable from those of flowering plants. The related linear Y-chain has been found in *A. thaliana* roots and the phylogenetic analysis of the galacturonosyltransferase that creates the Y-chain showed that it, AtXUT1, is a GT47 clade A protein belonging to a subclade different from MUR3 and with several *Physcomitrella* members[56], suggesting that these GalA-containing XyG sidechains are homologous rather than results of convergent evolution. Knocking out the galacturonosyl-transferase in *A. thaliana* resulted in shorter root hairs, suggesting a role of AtXUT1 and GalA containing XyG in root hair elongation[56]. Mutations in the XyG-specific galacturonosyl-transferase (MpXUT1) gene in the liverwort *M. polymorpha* resulted in very short rhizoids that eventually burst, suggesting a role of the GalA containing XyG in the tip-growing rhizoids[62], but the exact function is still unclear. These GT47 members belong to clade A of GT47 and all transfer glycosyl residues to the *O*-2 position of xylose in XyG.

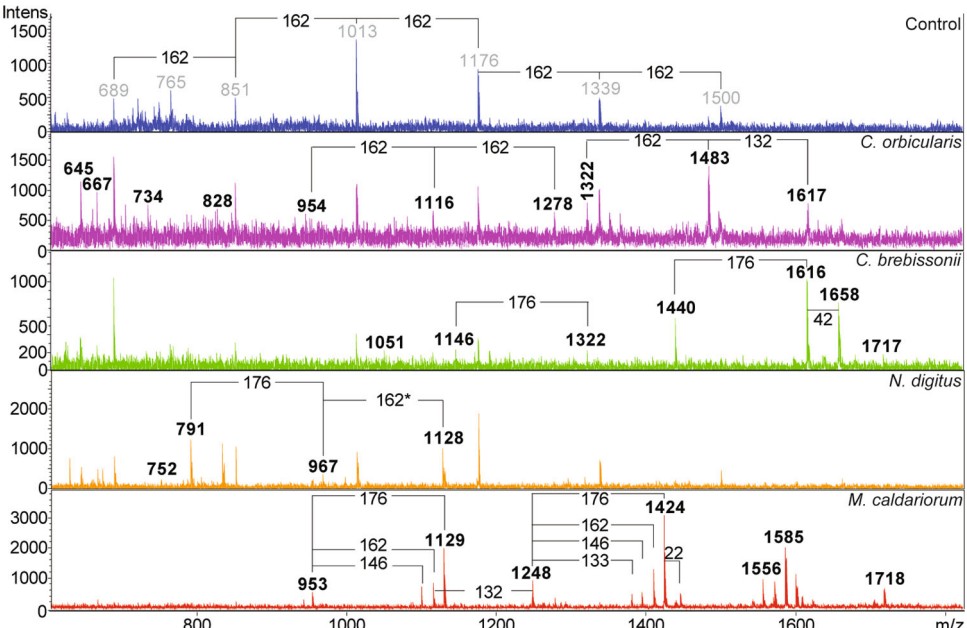

**Fig. 6 Maldi-ToF on xyloglucanase treated *Netrium digitus, Mesotaenium caldariorum* and *Cylindrocystis brebissonii* shows xyloglucanase released oligosaccharides.** Blue: *Coleochaete orbicularis* before treatment with xyloglucanase, the CGA all contain related contaminating hexose oligosaccharide compounds shown here with *m/z*. Purple: *Coleochaete orbicularis* after xyloglucanase treatment shows release of XGOs with mass differences corresponding to hexose (Glc/Gal) and pentose (Xyl). Green: xyloglucanase also released XGOs from *Cylindrocystis brebissonii* with mass differences corresponding to uronic acid (GalA/GlcA) and a possible XyG acetylation. Orange: XGOs released from *N. digitus* show mass differences of hexose as well as uronic acid. Red: XGOs released from *Mesotaenium caldariorum*, including mass differences corresponding to hexoses, pentoses, uronic acids as well as possible fucoses.

The F-chain has been found in hornworts but is absent from mosses. This absence is corroborated by sequence comparisons of moss GT37 proteins with those of vascular plants supporting the conclusion, that the clade responsible for L-chain fucosylation has no moss taxa[27]. Clade structure analysis is not without complications in family GT37[40], yet available evidence suggests that XyG fucosylation is lost in mosses; and there may be little selection pressure for preserving XyG fucosylation due to the type of conductive tissues in mosses[12]. The functional relevance of fucosylation to vasculature has been demonstrated by Brennan et al.[63] in an investigation of cell type specific occurrence of fucosylated XyG in commelinid and non-commelinid monocots. The authors found that XyG is not fucosylated in most tissues of commelinid monocots. However, with few exceptions, they were able to detect fucosylated XyG in cells associated with the vasculature in a range of commelinids. A similar investigation would be highly relevant to Solanales, which also generally feature non-fucosylated XyG. In tomato (Solanaceae) two XLT2 galactosyltransferases were found to be neofunctionalised to transfer L-Ara*f* rather than D-Gal*p* thus generating an S-chain rather than an L-chain[64] and at the same time eliminating the F-chain. However, synthesis of fucosylated XyG has been demonstrated in pollen tubes of tobacco[58], also belonging to Solanaceae, corroborating that fucosylation was not acquired multiple times during evolution but rather lost in tissues where no selection pressure favors its conservation. It appears that fucosylated XyG has functional properties that sets it apart from XyG in general[65] but it is not yet clear what precisely is the decisive property.

## Methods

**Bioinformatics**. Genome sequences from the CGA *Mesostigma viride* and *Chlorokybus atmophyticus*[10], *Chara braunii*[35], *Klebsormidium flaccidum*[34], *Spirogloea muscicola* gen. nov., *Mesotaenium endlicherianum*[36], and CGA transcriptomic data was obtained[40,66] from 1KP[33]. In short a local BLAST database consisting of the glycosyltransferases of CAZy[67] (http://www.cazy.org) with the addition of the glycosyltransferases from the land plants *A. thaliana, Oryza sativa, S. moellendorffii,* and *P. patens*, acquired from Genbank, using Harholt et al.[40] as guide, was generated. The proteome of each taxon was blasted against this data with a cut-off of $10^{-25}$ and quality controlled[66]. Phylogenetic analysis was done as in Harholt et al.[40] using Phylogeny.fr (http://phylogeny.lirmm.fr/phylo_cgi/index.cgi)[68]. *Penium margaritaceum* GT47 phylogenies were estimated by alignment using Muscle[69]; trimming in TrimAl[70] with gt=0.05, and trees were generated using PhyML[71]. Keys to trees are provided in Supplementary data 1.

**Cloning of Coleochaete orbicularis genes**. Cells from a fresh culture of *Coleochaete orbicularis* were harvested by centrifugation at 4000×g, resuspended in 2 mg/mL Driselase (Sigma-Aldrich) and incubated with gentle shaking for 30 min at room temperature. Protoplasts were harvested at 4000×g. App. 0.5 g cells were transferred to a mortar, ground in liquid nitrogen and RNA was extracted with Spectrum Plant Total RNA kit (Sigma-Aldrich). cDNA was synthesized with SuperScript III First strand SuperMix (Invitrogen).

Two putative and partial *Coleochaete orbicularis* GT47A sequences of 447 and 661 bp were identified in the 1 Kp data set by alignment to land plant sequences (*Selaginella, Physcomitrella, Amborella, Vitis* and *Cucumis*). Primers were constructed for the sequences (FW1 and RV; Table S1) and PCR using LA Taq polymerase (Takara) was performed on *Coleochaete orbicularis* cDNA. PCR resulted in gene fragments of the expected sizes and were confirmed by sequencing.

To obtain the 5′ end of the *CoGT47A1* gene, a forward primer (FW2) was placed in the conserved regions in the 5′ end of a *Coleochaete scutata* sequence (found in the 1KP dataset: 10200_VQBJ_GT47_MUR3/1409). This primer was used with the *CoGT47A1* reverse (RV) primer and resulted in the expected product and was cloned and sequenced and found to have perfect overlap with the *CoGT47A1* FW + RV PCR product.

The 3′ end of *CoGT47A1* was amplified by a nested PCR using first *CoGT47A1* FW3n and an anchored oligo-dT primer followed by nested PCR reaction *CoGT47A1* FW4n and anchored oligo-dT primer. The obtained product where cloned and sequenced.

The 3′ end of *CoGT47A2* was amplified by a nested PCR first using *CoGT47A2* FW1 and an anchored oligo-dT primer. After purification of the PCR product, another PCR was performed using *CoGT47A2* FW2 and RV, which resulted in a band of approximately 1.3 kb. The fragment was cloned and sequenced. The 1.3 kb fragment was 1319 nucleotides long excluding the poly(A) tail.

To obtain the 5′ end of both *CoGT47A* genes, 5′ RACE was performed using SMARTer RACE5′/3′ kit (ClonTech/Takara) according to manufacturer's instructions using the GSP1 and NGSP1 primers for the respective genes (Table S1). The resulting PCR products were cloned and sequenced. The full-length coding sequences were amplified by PCR using the primers FLFW and FLRV for

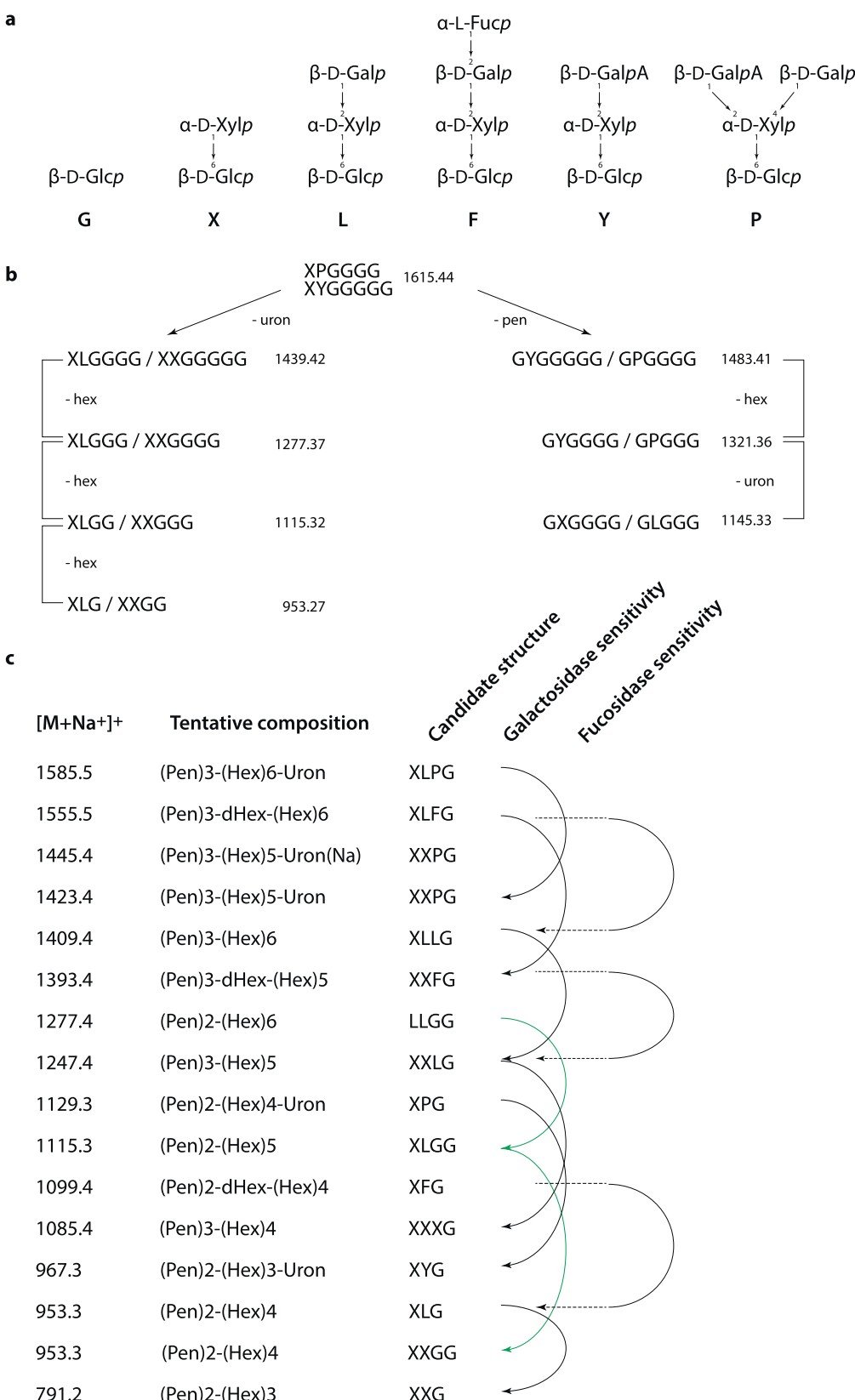

**Fig. 7 Tentative assignments of peaks in *Coleochaete orbicularis* and *Cylindrocystis brebissonii* and suggested XyG structures found in *Mesotaenium caldariorum*. a** Relevant XyG structures including standard one-letter codes for the XyG sidechains represented in **b, c** sodium adducts. **b** Putative-related XGOs are released after xyloglucanase treatment in *Coleochaete orbicularis* and *Cylindrocystis brebissonii*. Theoretical masses are indicated here while Fig. 6 reports recorded masses. **c** Tentative monosaccharide composition and candidate structures in *Mesotaenium caldariorum* are provided along with the enzyme sensitivities of the proposed oligosaccharides. Pen pentose, Hex, hexose, dHex deoxy-hexose, Uron uronic acid.

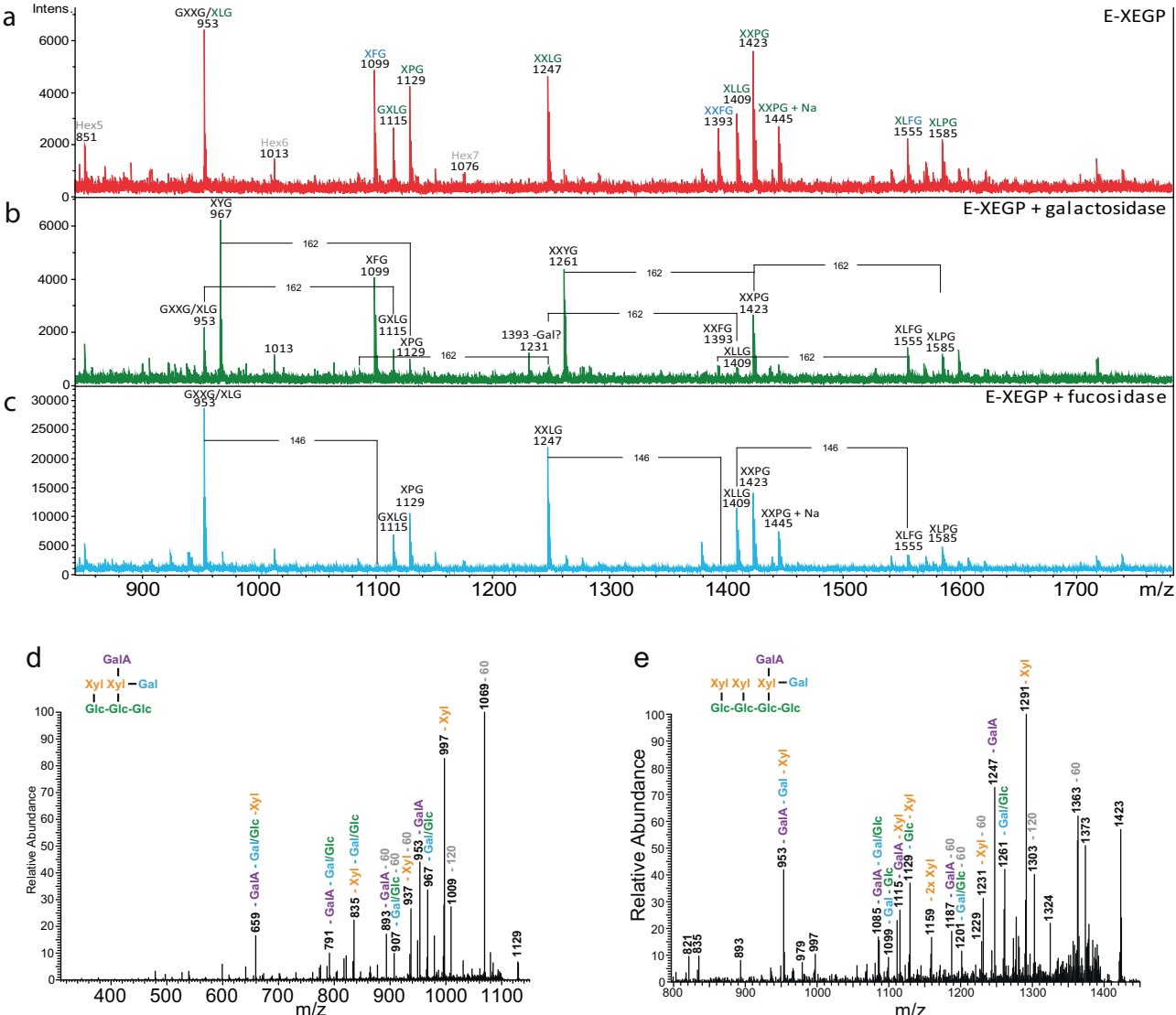

**Fig. 8 Maldi-ToF and tandem mass spectrometry data on *Mesotaenium caldariorum* treated with different XyG specific enzymes shows fucosylated and galacturonic acid containing XyG. a** Xyloglucanase treated *Mesotaenium caldariorum* results in release of many XGOs, corresponding to a core structure of primarily XXXG but also smaller fractions of XXG. **b** Degradation by xyloglucanase followed by degradation of galactosidase, degradable masses are indicated by lines including the mass of galactose (162). **c** Degradation by xyloglucanase followed by treatment with fucosidase, degradable masses are indicated by lines including the mass of fucose (146). Tandem mass spectroscopy of mass 1129 and 1423 show in **d** the loss of a pentose from *m/z* 1129, indicating a terminal pentose, most likely a xylose. **e** Loss of two consecutive pentoses from the *m/z* 1423 XGO is yielding an *m/z* 1159 ion. This suggests that the structures of the XGOs in **d** and **e** are related compounds and that the likely structure is XPG/PXG and XXPG/XPXG/PXXG for 1129 and 1423, respectively. Cross ring cleavage was also observed and is indicated with loss of *m/z* of 60 and 120 from the parent ion. The decoration of xyloglucan is normally more elaborate nearing the unsubstituted glucose, so although the most decorated sidechain may reside on other glucoses, we have for simplicity drawn the structure with the most decorated sidechains furthest towards the undecorated glucose.

CoGT47A1 and CoGT47A2, respectively (Table S1). The full-length genes were uploaded to Genbank with the following accession numbers: MW149251 for *CoGT47A1* and MW149252 for *CoGT47A2*.

**Plant materials, cultures, and media**. An *Anthoceros caucasicus* (hornwort) cell-culture was kindly donated by Dr Maike Petersen, University of Marburg, Germany; rose, spinach, maize cell-cultures were as reported by Popper and Fry[72]. CGA for CoMPP and TLC were grown for 10–14 days and harvested by centrifugation 800×*g* for 1 min[13]. *Chara* was grown at the Skidmore college Greenhouse wet wall. The CGA for MALDI-ToF analysis were grown as follows: *Netrium digitus* (LB599 University of Texas Algal Culture Collection or UTEX culture collection) and *Cylindrocystis brebissonii* (UTEX 1922) were cultured in 250 mL flasks containing 125 mL of sterile WHM medium supplemented with soil extract (10% final concentration) at 20 °C, 16 h light/8 h dark photocycle with 2000 lux of cool white fluorescent light, *Coleochaete orbicularis* (UTEX LB2651) was grown in petri dishes containing sterile liquid WHM supplemented with of peat extract (5%

final concentration) under the conditions described above but with the temperature reduced to 18 °C. *Mesotaenium caldariorum* (UTEX 41) was grown 10–14 days in WHM medium. All CGA material wash with fresh sterile medium and frozen in liquid $N_2$ and lyophilized.

**Preparation of AIR from CGA**. Lyophilized CGA samples were ground using liquid $N_2$ and agate mortar. Seventy percentage of EtOH was added 3–5 times and incubated RT for 10 min and 100 rpm. When no further green color was extracted from the EtOH extraction, 96% EtOH was added and incubated at RT for 10 min at 100 rpm. chloroform:methanol was added in an 1:1 ratio and 100 rpm for 10 min. the sample was centrifuged 10 min at 4000 rpm after each treatment and the supernatant discarded. Acetone was added in five volumes and incubated 5 min at 100 rpm. Samples were dried in the fume hood at RT overnight.

**CoMPP**. Different monoclonal antibodies were used to probe CoMPP arrays: LM15[54], LM25[49] (purchased from Plantprobes) and the CCRC-M1 and

CCRC-M39 (purchased from CarboSource). The antibodies CCRC-M1[73] and M39[74] are both specific towards fucosylated XyG.

Algal AIR was extracted with EDTA and NaOH, printed on arrays and blocked with skim milk solution[47,49]. After blocking, arrays were treated with enzymes; xyloglucanase E-XEGP (GH5) from *Paenibacillus* sp. (Megazyme, Bray, County Wicklow, Ireland) in 100 mM sodium acetate buffer pH 5.5 at concentrations of 1.0 U/mL or MFuc5 in 20 μM citrate-phosphate buffer pH 6 at concentrations of 0.02 mg/mL[59] and incubated overnight at 30 °C and 80 rpm. When treating the arrays with the xyloglucanase XcXGHA[48] a concentration of 0.5 mg/mL was used in 100 mM sodium acetate buffer pH 6.5 and incubated overnight at 40 °C and 80 rpm. Arrays were washed three times and probed with antibodies and developed[49]. The complete data set is provided in Supplementary data 1.

**Immunolabelling of CGA samples**. For live cell labeling, 5-day old cultures of *Netrium digitus*, *Cylindrocystis brebissonii* and *Spirogyra communis* were collected in 15 mL centrifuge tubes and centrifuged on an IEC clinical centrifuge at 1000×*g* for 1 min. The supernatant was removed and the cells were washed with fresh growth medium. This washing was repeated 3×. The *Coleochaete* species were collected after 2 weeks of growth by gentle scraping of thalli off the bottom of petri dishes and placing in 15 mL centrifuge tubes. The thalli were washed as described above. Washed algal samples were first blocked for 30 min with gentle shaking at room temp in 0.5% Carnation Instant non-fat milk dissolved in growth medium. The samples were then washed with fresh growth medium as described above, and then incubated in primary antibody solution consisting of a 1/10 dilution of LM15 in fresh growth medium. The samples were gently shaken on a laboratory shaker for 90 min in the dark. The samples were then washed 3× with growth medium, blocked as described above for 30 min and washed again. They were then incubated for 90 min with gentle shaking in the dark with secondary antibody consisting of a 1/50 dilution of anti-rat TRITC (Sigma Chemical, St. Louis, MO, USA) in growth medium. The samples were washed 3× with growth medium and kept in the dark until observation. The control treatment was as above but included elimination of primary antibody.

*Mesotaenium caldariorum* cells were grown as described above. For cell cycle synchronization, cells were collected by centrifugation after 10–14 days of growth and kept in the dark for 2 weeks at 20 °C, followed by washing in WHM media and reintroduction into the normal day/night cycle.

*Chara corallina* culturing, fixation, sectioning and immunolabeling followed the protocol as described in Domozych et al.[30]. All samples were viewed with an Olympus BX-61 microscope equipped with an Olympus Fluoview 300 confocal unit. A green laser with TRITC filter set was employed for viewing fluorescence. DIC imaging employed both blue and green lasers. A ×100 oil emission magnifying objective was used, and excitation wavelengths used were 488 nm for chlorophyll autofluorescence and 555 nm for TRITC.

**TLC analysis**. Merck silica gel 60 TLC plates and solvents were obtained from VWR. TLC was performed as as in Franková and Fry[75] with thymol staining according to Jork et al.[76]. AIR samples of the CGA species *Netrium digitus*, *Coleochaete orbicularis*, and *Mesotaenium caldariorum* as well as AIRs of cell cultures of *Anthoceros* (hornwort), rose, spinach, and maize were used. The AIR was freed of any ester groups (acetyl etc.) by saponification in 1 M NaOH at 20 °C for 30 min, which was then adjusted to pH 4.7 with acetic acid and any solubilized polysaccharides were re-pelleted in ethanol. One portion of the saponified AIR was treated with 1% (w/v) Driselase in PyAW (pyridine/acetic acid/water, 1:1:98, pH 4.7, containing 0.5% chlorobutanol) at 20 °C for 3 days, then only the ethanol-soluble digestion products (principally mono- and disaccharides) were resolved by TLC on silica-gel in butan-1-ol/acetic acid/water 4:1:1 (two ascents) and stained with thymol/$H_2SO_4$. Additional portions of AIR were treated with XEG (0.01% XEG in PyAW, or enzyme-free controls) at 20 °C for 15 min, and only the ethanol-soluble digestion products (oligosaccharides) were collected: these were treated with human salivary α-amylase, resolved by TLC on silica-gel in butan-1-ol/acetic acid/water 2:1:1 (two ascents), and stained with thymol/$H_2SO_4$.

**Enzymatic digestion of CGA AIR for MALDI-ToF and tandem MS analysis**. CGA AIR samples (2.5 mg per experiment) were washed three times in ice-cold water, by vortexing and spinning at 15,000 rpm for 10 min. Buffer (5 mM sodium acetate buffer pH 6) was added and the mixture was incubated 10 min at RT, vortexed vigorously and spun 10 min at 10,000 rpm. The supernatant was discarded, leaving only a small amount of fluid over the pellet. Buffer was added 5 mM sodium acetate buffer pH 6 with or without xyloglucanase E-XEGP 0.1 U/mL and incubated at 40 °C overnight and 1000 rpm. Samples were vortexed vigorously and spun 10 min at 10,000 rpm. The supernatants were collected and spun additionally for 10 min at 10,000 rpm resulting in the XGO mixture. Further enzymatic degradation with fucosidase MFuc5 or galactosidase (Megazyme; β-galactosidase E-BGLAN from *Aspergillus niger*) was performed by addition of 0.2 μg/mL of MFuc5 from *Anaerolinea thermophila* or 16 U/mL galactosidase to the XGO mixture and incubation overnight at 30° and 100 rpm.

**MALDI-ToF and tandem MS analysis**. Two microliter of a 9 mg mL$^{-1}$ solution of 2,5-dihydroxybenzoic acid (DHB) in 30% (v/v) acetonitrile was applied on a target plate (MTP 384 target plate ground steel TF from Bruker Daltonics). One microliter sample was then mixed into the DHB droplet followed by drying under a stream of hot air. The analysis was run on an Ultraflex MALDI-ToF-ToF (Bruker Daltonics, GmbH, Bremen, Germany) with a Nitrogen 337 nm laser beam. The instrument was operated in positive acquisition mode and controlled by the FlexControl 3.0.184.0 software package. The acquired spectra were analyzed on FlexAnalysis software (version 3.0.96.0). The acquisition range used was from *m/z* 200 to 4000. The data were collected from averaging 1000 laser shots, with the lowest laser energy necessary to obtain sufficient signal to noise ratios.

**Mass spectrometry of XyG oligo-saccharides, direct injection MS**. For direct injection, oligo-saccharides were analyzed using an LTQ-Velos Pro linear ion trap mass spectrometer (Thermo Scientific, San Jose, CA, USA) connected to an Ultimate 3000RS HPLC (Dionex, Sunnyvale, CA, USA). The setup was used for direct injection without a column; the pump delivered 200 μL min$^{-1}$ 70% acetonitrile containing 1 mM formic acid, and data was acquired for 24 s after injection. For the MS, the capillary voltage was set to 3.5 kV and the scan range was *m/z* 800–2000 using two micro scans. The automatic gain control was set to 10,000 charges and the maximum injection time was 20 ms. For fragmentation of selected precursor ions by MS/MS, the normalized collision energy was set to 35 and two-four micro scans were used.

**Statistics and reproducibility**. Three extractions were performed for CoMPP analysis each spotted in four dilutions resulting in 12 spots per sample. CoMPP is a semiquantitative analysis and suited for assessing relative differences only, hence the choice of heatmaps for data presentation. Biological variability, where observed, is discussed in the main text.

**Reporting summary**. Further information on research design is available in the Nature Research Reporting Summary linked to this article.

## Data availability

The data that support the findings of this study are available from the corresponding author upon reasonable request. Sequence data that support the findings of this study have been deposited in Genbank MW149251 for *CoGT47A1* and MW149252 for *CoGT47A2*.

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

## Acknowledgements
J.H. was supported by the Carlsberg Foundation.

## Author contributions
M.D.M., J.H., B.W., D.D., S.C.F., I.E.J., J.U.F., M.Ł., T.F., L.N., and P.U.: Performed all experiments and analyzed results. M.D.M., J.H., P.U., S.C.F., and D.D.: Data visualization, data curation and formal analysis. W.G.T.W., P.U., and M.D.M.: Project administration. W.G.T.W., P.U., J.H., and J.D.M.: Experimental ideas and supervision M.D.M., J.H., and P.U.: Paper writing with help from all co-authors.

## Competing interests
The authors declare no competing interests.
