## [Peer Review File · Communications Biology]

Reviewers' comments:

Reviewer #1 (Remarks to the Author):

The manuscript by Maria Dalgaard Mikkelsen et al. entitled "Ancient Origin for fucosylated xyloglucan in charophycean green algae" demonstrates with some certainty, that the cell wall polysaccharide xyloglucan has its origin prior to terrestrialization, a statement which was so far still under debate. The authors provide biochemical and structural evidences of xyloglucan structures in a series of charophycean green algae (CGA). They also indicate the occurrence of some substitutions (notably fucosylation) on xyloglucan motifs in CGA, indicating that these structural specificities are not restricted to late evolution events in land plants. This has implications on our understanding of the origin and the evolution of xyloglucan across the green lineage, including regarding the biosynthetic enzymes involved and its functional roles in planta. The authors used a compelling series of experimental approaches, from gene analyses, to xyloglucan identification (in extracts and within algal cells) and structural analyses. This is a nicely executed piece of work, presented in a concise way. The structure and the writing style are excellent. Yet, I have some comments detailed below that would deserve consideration prior to publication.

Main comment

The authors should carefully check that all figures and tables are correctly referenced within the main text. I also have some concern with the phylogenetic trees where it is sometimes hard to distinguish between the green colours, and this may be a problem for readers with impaired colour vision, especially if you look for a specific target. I would suggest to change colours and to highlight in figures (with a bold police?) the sequences which are cited within the manuscript.

More specifically:

Page 3 line 68. The Spirogloea sequence(s) cited within the text is not shown on tree (Figure2) –or the name/acronym not apparent

Page 3 line 80. Table S1 refers to CoMPP analysis –I guess FigS2 should be mentioned instead (or Suppl File1). Also it might be useful to indicate where these Penium sequences are located in the tree (as they are discussed in the text), I guess they stand within one of the CGA group shown. Figure S2. Specify the location of the cloned Co sequences on tree (I could not find CoGT47A2).

Page 4 line 93. There is only one table in the main paper (no Table 2) –I guess this should refer to Table1

Figure 4 seems to have been cropped during the pdf conversion process.

Other comments

Introduction. Please state the letter nomenclature for xyloglucan within the introduction. At present the letters are apparent in FigureS2 but only explained at the end of manuscript. This should appear prior to Figure 7a.

Page 3, line 71. All these enzymes have not been functionally characterized yet in CGA. Replace 'have a full biosynthetic inventory for production' by "may have a full biosynthetic inventory for production"

Page 3, line 73. The situation of the GT34 family is not really discussed as compared to GT47/37. The Mes sequences seem to be rather scattered along the tree. One sentence commenting those would be appreciated.

Page 3 line 74. "We clone two full-length C.o. sequences". Please specify for bioinformatics/sequences analyses. At first read I was anticipating a cloning for functional studies, while the sequences are "only" used (as far as I understand) for the bioinformatics/phylogenetic analyses. This would deserve a bit more of context/explanation. Are functional studies foreseen?

Page 4 line 93. Briefly introduced the antibodies specificities. Or refer to the Mat&Met section.

Page 4 line 97. Specify a XEGP treatment

Page 4, lines 96-98. I cannot figure out why a LM15 binding would be recalcitrant to the enzymatic treatments and not LM25, especially as the authors tried another enzyme in addition to the classical one. Are additional decorations/interactions foreseen (very unlikely) and impairing the enzymatic degradations? But I guess one would not expect those antibodies to bind in this case. Do the authors have any explanation on this? –the dynamic option is not the only answer to this I think.

Page 4 line 117. Replace by "epitopes were detected"

TableS3. Have the CCRC-M1/M39 antibodies been used with *M. caldariorum* only? This would have been interesting to screen the full CGA dataset.

Figures 3,4. Are the labelling results with LM15 correlating with any LM25 labelling?

Figure 5 and Table1. This might have been useful to show in Table 1 the same species screened for structural analyses in Figure5 (presently apparent in Table S1)

Page 6 lines 165-174. Better place in discussion.

Page 11, lines 332, 334. Mind °C

Figure1. Please expand " v p" –I guess vascular plants

Reviewer #2 (Remarks to the Author):

This work provides several lines of evidence that plant xyloglucans evolved within the charophycean green algae. The authors have used recently published data to identify xyloglucan orthologs and produced phylogenies that show these orthologs in context with plant xyloglucans. Microarray data have been expanded to include more taxa and more individuals and labeled with two antibodies to xyloglucans before and after treatment with xyloglucanase. The structure of these xyloglucans has been elucidated.

This work will be of interest to many researchers. A case has been strengthened for presence of xyloglucans at different developmental stages associated with growth or for masking of xyloglucans by other cell wall molecules. I think the manuscript is in good shape, though problems with interpretation or labeling of some of the micrographs need to be addressed.

major comments

1. p4 lines 95-96 Which two species of late divergent taxa do not label? Based on Table 1, it would seem to be *Pleurotaenium trabecula* & *Teilingia granulata*, but in Table S1, both of these have labeling and several other species do not. I think this needs to be clarified or this statement left out.

2. Captions for Fig 3

3a. Zygote is not the right word. This young thallus has grown from a scaly zoospore that settled down, developed a cell wall and divided to form the thallus. It is the remnants of the zoospore wall that appear to have labeled. This result fits with the idea presented in the manuscript that newly formed walls label with LM15. Labeling is also seen at the thallus periphery, where growth occurs.

3b. The body form of *Coleochaete* is a thallus, not a colony. I would assert that these are hairs, rather than "hair-like structures". Hence suggest changing to say "The peripheral cells of thalli as well as elongating hairs also show distinct labeling."

3c. I'm not convinced that there is strong labeling at "the tips of hair-like structures". Instead I see what appear to be two mature hairs parallel to each other (at top of DIC image pointing to 11 o'clock) that are NOT labeled, and a big green labeled blob below that appears to be at the end of a growing filament.

3. Fig 4 lettering

I believe that the letters on the images have been misplaced. Instead of labels across the top row and then the bottom row, I think that the image pairs go together. Hence the top row should be labeled a, c, e, g and the bottom row b, d, f, h. If so, the text in the body of the paper and the figure caption would make sense. The parenthetical remark in the caption for g-h is confusing, however. Perhaps you could point out the chlorophyll with an arrow. I can see what appears to be chlorophyll in the second image from left on top row, but not here.

minor comments

p3 line 70 Orthologs...were also

p3 line 80 Table S1 doesn't seem to be the correct reference. Do you mean Fig S2?

p3 line 82 Schultink et al. 2014 is ref 24

p3 line 84 as well as the notion

p4 line 93 Table 2 is referred to, but there is no Table 2. Do you mean Table 1?

In Table S1 the specific epithets of *Coleochaete scutata* and *Coleochaete orbicularis* are misspelled. Do you want to state what the difference is between Table 1 and Table S1? The titles are pretty much identical. Is there a reason why the data for LM15 in Table 1 are presented with NaOH before CDTA, in contrast to Table S1 and in contrast to the data for LM25 in both Table 1 and Table S1?

p4 line 117 epitopes were detected?

p9 line 269 Driselase is misspelled.

In the gene phylogeny figures (Fig 2 and Figs S1-S3) it is difficult to distinguish between the colors representing *Selaginella* & *Physcomitrella*.

In caption for Fig S7, devoid of chlorophyll (not diploid).

In caption for Fig S8, instead of blue line should it say purple?

The references are sometimes listed twice with slightly different formats (for example 12 & 27, 13 & 71)

April 20, 2021

Dear Reviewers,

The feedback to our manuscript is greatly appreciated. We find the critique constructive and are grateful for your help with weeding out errors and mistakes. We believe that we have been able to address all the issues that you have presented us with.

Before presenting our handling of each of your points, there are two commonalities:

Firstly, the xyloglucanase sensitivity or lack thereof in relation to the differences between the two antibodies LM15 and LM25. Obviously, we have been pondering the implications of this but did not include our thoughts in the first version thinking that they were unavoidably speculative. But it is correct that reader would want to know our thinking and in the new version we have offered an explanation, added a new citation of an analogous phenomenon seen previously and we provide more information on the difference between the two antibodies.

Secondly, there is the color coding of the phylogenetic trees. We did in fact try to be conscientious about color blind readers by not including green and red in the same figure as red/green color blindness is the most common. However, we have now increased the contrast of the color coding in the phylogenetic trees while maintaining a green to blue color scheme, from flowering terrestrial plants to marine prasinophytes.

Then there are actions taken in response to issues raised by each reviewer.

Reviewer #1

The authors should carefully check that all figures and tables are correctly referenced within the main text.

Done

The *Spirogloea* sequence(s) cited within the text is not shown on tree (Figure2)

The numbers referring to the *Spirogloea* sequences are now given both in relation to figure 2 and Fig S1. References to *Penium* sequences have been clarified as well.

Page 3 line 80. Table S1 refers to CoMPP analysis –I guess FigS2 should be mentioned instead (or Suppl File1).

Correct, we have changed it to refer to Fig S2

There is only one table in the main paper (no Table 2) –I guess this should refer to Table1

Yes, that has been corrected.

Specify the location of the cloned Co sequences on tree (I could not find CoGT47A2).

Correct – it belongs in the tree in the same location as CoGT47A1. We have now indicated in the figure even though it is listed among highly similar sequences in Suppl. File 1.

Introduction. Please state the letter nomenclature for xyloglucan within the introduction.

Done

All these enzymes have not been functionally characterized yet in CGA. Replace 'have a full biosynthetic inventory for production' by "may have a full biosynthetic inventory for production"

We agree and have substituted "demonstrating" with "suggesting"

The situation of the GT34 family is not really discussed.

We agree, GT34 is enigmatic and deserves discussion. We have expanded the coverage of GT34.

"We clone two full-length C.o. sequences". Please specify for bioinformatics/sequences analyses. At first read I was anticipating a cloning for functional studies, while the sequences are "only" used (as far as I understand) for the bioinformatics/phylogenetic analyses. This would deserve a bit more of context/explanation.

Coleochaete often take center stage in discussions in the literature of terrestrialization. If *Coleochaete* holds a particularly interesting evolutionary position, then we would like not only to rely on the transcripts found in the 1Kp project (although they are often long and of very good quality). This was the rationale for cloning the *Coleochaete* XLT2-like sequences. They turned out not to add new information to the phylogeny but to strongly corroborate what the 1Kp sequences already taught us. We have added a sentence to clarify this.

Page 4 line 117. Replace by "epitopes were detected"

It has been corrected

TableS3. Have the CCRC-M1/M39 antibodies been used with *M. caldariorum* only. We have indeed performed the experiments with the other species than we go more into detail with in the current paper. They have been added to the Table S3 and the sentence in the text has been slightly rephrased.

Figure 5 and Table1. This might have been useful to show in Table 1 the same species screened for structural analyses in Figure 5 (presently apparent in Table S1)

We agree, however, Maldi-tof analysis requires that the XyG is present in larger quantities and that it can be degraded by xyloglucanases to a measurable extent. This is unfortunately not the case for many of the species with the currently available xyloglucanases. We have added a sentence in the text explaining our selection criteria.

Page 6 lines 165-174. Better place in discussion

Done.

Page 11, lines 332, 334. Mind °C

Done

Figure1. Please expand " v p" –I guess vascular plants

The figure was unfortunately cropped during PDF formatting, this has now been corrected.

Reviewer #2

p4 lines 95-96 Which two species of late divergent taxa do not label? Based on Table 1, it would seem to be *Pleurotaenium trabecula* & *Teilingia granulata*, but in Table S1, both of these have labeling and several other species do not. I think this needs to be clarified or this statement left out.

We agree that this was not clearly written in the text. We have rephrased the text.

Captions for Fig 3

3a. Zygote is not the right word. This young thallus has grown from a scaly zoospore that settled down, developed a cell wall and divided to form the thallus. It is the remnants of the zoospore wall that appear to have labeled. This result fits with the idea presented in the manuscript that newly formed walls label with LM15. Labeling is also seen at the thallus periphery, where growth occurs.

We thank the reviewer for this valuable information and clarification. We have corrected this in the figure legend.

The body form of Coleochaete is a thallus, not a colony. I would assert that these are hairs, rather than "hair-like structures". Hence suggest changing to say "The peripheral cells of thalli as well as elongating hairs also show distinct labeling."

We are grateful for this valuable correction, which we have inserted

I'm not convinced that there is strong labeling at "the tips of hair-like structures". Instead I see what appear to be two mature hairs parallel to each other (at top of DIC image pointing to 11 o'clock) that are NOT labeled, and a big green labeled blob below that appears to be at the end of a growing filament.

This was a mistake and we thank the reviewer for catching it. We have corrected the "hair-like" structures to "growing filament".

Fig 4 lettering

I believe that the letters on the images have been misplaced. Instead of labels across the top row and then the bottom row, I think that the image pairs go together. Hence the top row should be labeled a, c e, g and the bottom row b, d, f, h. If so, the text in the body of the paper and the figure caption would make sense. The parenthetical remark in the caption for g-h is confusing, however. Perhaps you could point out the chlorophyll with an arrow. I can see what appears to be chlorophyll in the second image from left on top row, but not here.

We agree that lettering was wrong as suggested by the reviewer and we have changed them accordingly. The parenthetical remark has been moved to the right position and arrows have been inserted in the figure to indicate the position of the chlorophyll

p3 line 70 Orthologs...were also

Done

p3 line 80 Table S1 doesn't seem to be the correct reference. Do you mean Fig S2?

This has been corrected

p3 line 82 Schultink et al. 2014 is ref 24

this has been corrected

p3 line 84 as well as the notion

“the” has been inserted

p4 line 93 Table 2 is referred to, but there is no Table 2. Do you mean Table 1?

Yes, the reference has been changed in the text

In Table S1 the specific epithets of *Coleochaete scutata* and *Coleochaete orbicularis* are misspelled. Do you want to state what the difference is between Table 1 and Table S1? The titles are pretty much identical.

The misspelled names have been corrected. The title of Table S1 has been refined to state the difference to Table 1.

Is there a reason why the data for LM15 in Table 1 are presented with NaOH before CDTA, in contrast to Table S1 and in contrast to the data for LM25 in both Table 1 and Table S1?

That was a mistake and the NaOH and EDTA has been swapped in Table 1, so all figures are uniformly presented

p4 line 117 epitopes were detected?

Done

p9 line 269 Driselase is misspelled.

It has been corrected

In caption for Fig S7, devoid of chlorophyll (not diploid).

It has been corrected

In caption for Fig S8, instead of blue line should it say purple?

The line is now truly blue

The references are sometimes listed twice with slightly different formats (for example 12 & 27, 13 & 71)

The references have been corrected and adjusted to the right formatting.

On behalf of the authors

Peter Ulvskov

REVIEWERS' COMMENTS:

Reviewer #1 (Remarks to the Author):

The authors have done a great job in revising this manuscript and answering all my comments and more. I have no more comments to be addressed prior to publication. Congratulations on a nice piece of work.

Reviewer #2 (Remarks to the Author):

The revised manuscript looks good. I would suggest one small change to the caption for Fig 3c. Instead of " at the tips of growing filament structures" I would suggest "at the tips of growing filaments."